# Successful Prenatal Treatment of Cardiac Rhabdomyoma in a Fetus with Tuberous Sclerosis

Joachim Carsten Will [1,*], Nina Siedentopf [2], Oliver Schmid [3], Teresa Mira Gruber [2], Wolfgang Henrich [2], Christoph Hertzberg [4] and Bernhard Weschke [5,*]

1   Department of Pediatric Cardiology, Charité University Medicine Berlin, 13353 Berlin, Germany
2   Department of Obstetrics, Charité University Medicine Berlin, 13353 Berlin, Germany;
    nina.siedentopf@charite.de (N.S.); teresa-mira.gruber@charite.de (T.M.G.);
    wolfgang.henrich@charite.de (W.H.)
3   Praxis für Pränataldiagnostik, Schloßstr. 2, 13507 Berlin, Germany; post@dr-schmid-berlin.de
4   Diagnose-und Behandlungszentrum für Kinder und Jugendliche, Vivantes Klinikum Neukölln,
    12351 Berlin, Germany; christoph.hertzberg@vivantes.de
5   Department of Pediatric Neurology and Center for Chronically Sick Children,
    Charité University Medicine Berlin, 13353 Berlin, Germany
*   Correspondence: achim.will@charite.de (J.C.W.); bernhard.weschke@charite.de (B.W.)

**Abstract:** Cardiac rhabdomyomas are a possible early manifestation of the Tuberous Sclerosis Complex (TSC). They often regress spontaneously but may grow and cause cardiac dysfunction, threatening the child's life. Treatment with rapalogs can stop the growth of these cardiac tumors and even make them shrink. Here, we present the case of a successful treatment of a cardiac rhabdomyoma in a fetus with TSC by administering sirolimus to the mother. The child's father carries a *TSC2* mutation and the family already had a child with TSC. After we confirmed the TSC diagnosis and growth of the tumor with impending heart failure, we started treatment at 27 weeks of gestation. Subsequently, the rhabdomyoma shrank and the ventricular function improved. The mother tolerated the treatment very well. Delivery was induced at 39 weeks and 1 day of gestation and proceeded without complications. The length, weight, and head circumference of the newborn were normal for the gestational age. Rapalog treatment was continued with everolimus. Metoprolol and vigabatrin were added because of ventricular preexcitation and epileptic discharges in the EEG, respectively. We provide the follow-up data on the child's development in her first two years of life and discuss the efficacy and safety of this treatment.

**Keywords:** tuberous sclerosis; rhabdomyoma; heart failure; fetal therapies; sirolimus; treatment outcome





## 1. Introduction

With an estimated incidence of around 1:6000 live births, the Tuberous Sclerosis Complex (TSC) is one of the more common "rare" diseases. It is marked by the appearance of multiple benign tumors, called hamartomas, in different organs, such as the skin, brain, kidneys, lung, and heart. It is caused by a heterozygous *TSC1* or *TSC2* mutation that, in combination with a "second hit", leads to a loss of the hamartin–tuberin complex and thereby to a constitutive activation of the mammalian target of rapamycin complex 1 (mTORC 1) effecting cell proliferation and tumorigenesis [1]. With a percentage of more than 60%, cardiac rhabdomyomas are the most frequent primary heart tumors in children [2,3]; among prenatally diagnosed cardiac tumors, their percentage may be even higher. In a study of 53 cases, the genetic analysis showed that most of the fetal rhabdomyomas were caused by a *TSC1* or *TSC2* mutation, and the presence of multiple rhabdomyomas was significantly associated with TSC [4].

Cardiac rhabdomyomas are an early manifestation of the Tuberous Sclerosis Complex in newborns and children up to the age of two years [5]. They tend to regress in most

cases, either already in utero or postnatally by apoptosis for to date unknown reasons [2,6]. It has been discussed that estrogens play some role in this process as they can activate mTORC1 as well as mTORC2 [7], and the growth or new appearance of rhabdomyomas has been observed mainly in female adolescents [8]. Nevertheless, until their spontaneous attrition, these tumors can cause cardiac dysfunction by outflow tract obstruction or cardiac dysrhythmia, even already in the fetus [9]. An adverse neonatal outcome has been significantly associated with tumor size ($\geq$20 mm diameter), fetal dysrhythmia, and the presence of hydrops fetalis [6]. Addressing this problem, there have been successful trials, albeit few in number, treating the fetus with TSC-related rhabdomyoma by treating the pregnant mother with rapamycin analogs (rapalogs), which act against the constitutive activation of mTORC1 [10–15]. Sirolimus is used as an immunosuppressive agent to prevent a host vs. graft reaction in transplant patients, but also in patients with *TSC1/2* mutation-associated disorders, e.g., for the treatment of renal angiomyolipomas and pulmonary lymphangioleiomyomatosis. It has also been used with some success in patients with vascular tumors and vascular malformations, including children with the Kasabach–Merritt phenomenon [16], and in congenital hyperinsulinism [17]. Everolimus, another rapalog, is also used for its antiproliferative effect, e.g., for the treatment of subependymal giant astrocytomas (SEGAs), and recently also as an antiepileptic drug in patients with TSC [18]. The use of rapalogs in cancer is however limited, one of the presumed reasons being that mTORC1 inhibition leads to its feedback activation via the PI3K pathway [7].

Here, we report the case of a successful treatment of a large, possibly life-threatening cardiac rhabdomyoma in a fetus by administering sirolimus to the mother. We also compare it to the eight cases published to date [10–15], and discuss the pros and possible cons of this treatment.

## 2. Case Description

In a healthy, 31-year-old pregnant woman, an ultrasound examination of her fetus in the 21st week of gestation (GW) revealed a cardiac rhabdomyoma sized $11.5 \times 7.9$ mm extending from the septal part of the right ventricle to the right atrium. There was stenosis and an insufficiency of the tricuspid valve and reduced contractility of the right ventricle. The echocardiography (GE Voluson E10, GE Healthcare, Chicago, IL, USA; Figure 1) at 25 weeks and 1 day of gestation confirmed these findings, showing a giant rhabdomyoma in the entire right atrial cavity ($21.5 \times 39.7$ mm), in the right ventricular cavity and the complete intraventricular septum ($18.5 \times 29.9$ mm), and in the left ventricular posterior wall. The tumor covered the tricuspid valve, leading to impaired ventricular filling with additional tricuspid insufficiency and right ventricular decompensation with mild–moderate pericardial effusion (Figure 1). Ventricular subependymal nodules were detected by fetal MRI (Figure 2), thereby confirming the diagnosis of TSC presumably caused by a *TSC2* mutation (c.5160T > G, p.Asn1720Lys) transmitted from the father. The family already had a 2-year-old daughter with TSC.

Considering the impending heart failure, therapy by sirolimus (Rapamune®, Pfizer Pharma GmbH) administered to the mother was started at 27 weeks and 0 days of gestation, starting with 4 mg/day and then adjusting the dose to achieve trough levels of 9–12 ng/mL (recommendation: D. Ebrahimi-Fakhari [14]).

Pre-treatment examinations of the mother revealed slight hypercholesterinemia (total cholesterol 250 mg/dL, LDL-cholesterol 163 mg/dL) and anemia (Ery 3.7/pL, hemoglobin 11.7 g/dL), not considered contra-indications to the treatment. Differentiation of lymphocytes showed low natural killer cells (0.05/nL) only. IgG antibodies against CMV and EBV were present. During the treatment, blood samples were taken every two weeks to check the sirolimus serum levels, hematologic parameters, liver enzymes, lipase, urea, creatinine, and lipid status. Every 6 weeks, CD4 cells were additionally measured to monitor the possible negative effects on lymphogenesis.

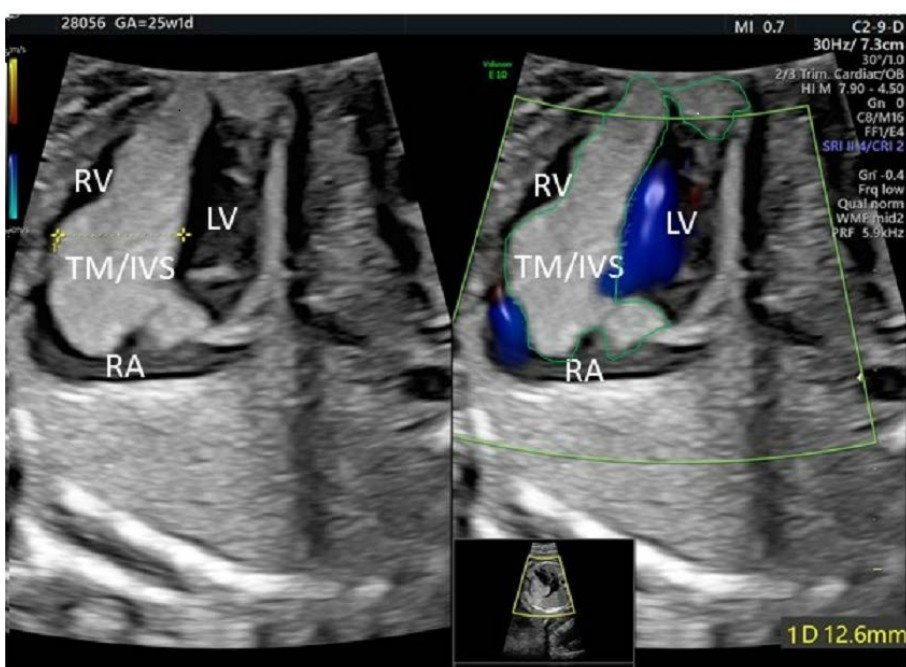

**Figure 1.** Echocardiographic four-chamber view at 25 + 1 GW showing a large rhabdomyoma covering nearly the entire right side of the cardiac structures. TM: tumor, IVS: interventricular septum; RA: right atrium; RV: right ventricle; LV: left ventricle (tumor circumference in green).

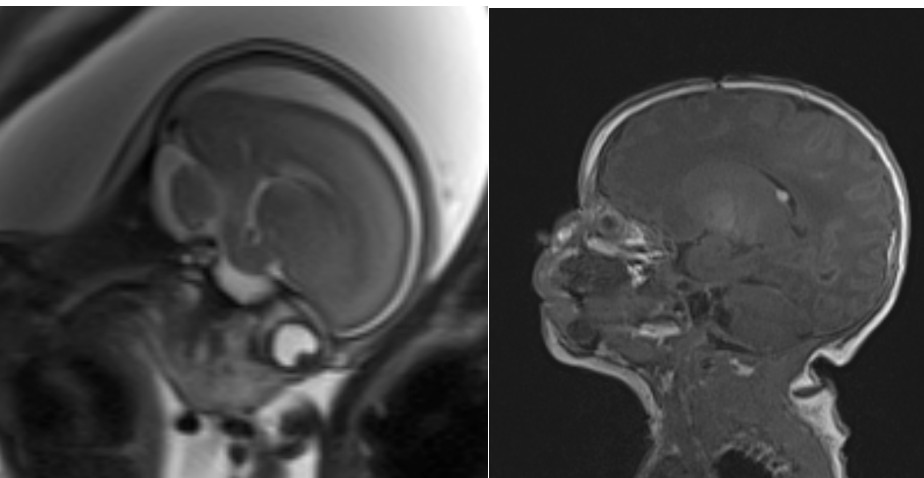

**Figure 2.** Left: fetal T2-weighed MRI at 24 gestational weeks. Right: postnatal T1-weighed MRI. Both show a subependymal nodule in the posterior right lateral ventricle.

With a dose of 4 mg/day (2.16 mg/m$^2$), sirolimus trough levels varied between 8.4–9.9 ng/mL and maximum levels (2 h after drug intake) did not exceed 14.2 ng/mL. In addition to high cholesterol levels that increased to max. 301 mg/dL and high triglycerides (max. 351 mg/dL), no substantial changes in blood values compared to pretreatment values were noticed. CD4 and CD8 cells remained in the normal range, and there was no reactivation of CMV as determined by PCR in the blood (Table 1). The mother tolerated the treatment very well and only reported small vesicles on her tongue once, which did not prompt any change in treatment. Echocardiography was performed every two weeks, showing a reduction in rhabdomyoma size and the amelioration of the ventricular function.

**Table 1.** Selected laboratory values of the mother. GA, gestational age; * at delivery; ** 8 days after terminating sirolimus treatment.

| Parameter | Unit | GA 27 Weeks | GA 31 Weeks | GA 37 Weeks | GA 39 Weeks * | Reference Values |
|---|---|---|---|---|---|---|
| CD4 cells | /nL | 0.92 | 0.96 | 0.97 | | 0.50–1.20 |
| CD8 cells | /nL | 0.33 | 0.38 | 0.37 | | 0.30–0.80 |
| CD4/CD8 | ratio | 2.8 | 2.5 | 2.6 | | 1.1–3.0 |
| NK cells | /nL | 0.05 | | | | 0.10–0.40 |
| Hemoglobin | g/dL | 11.8 | 11.3 | 11.1 | 10.2 | 12.0–15.6 |
| Erythrocytes | /pL | 3.7 | 3.5 | 3.7 | 3.4 | 3.9–5.2 |
| Leukocytes | /nL | 8.89 | 7.38 | 7.76 | 7.96 | 3.90–10.50 |
| Thrombocytes | /nL | 178 | 209 | 193 | 189 | 150–370 |
| Creatinin | mg/dL | 0.66 | | 0.59 | | 0.50–0.90 |
| GPT (ALT) | U/L | 9 | 15 | 20 | | <31 |
| GOT (AST) | U/L | 16 | 24 | 26 | | <35 |
| Cholesterin | mg/dL | 250 | 294 | 301 | | <200 |
| LDL-Cholesterin | mg/dL | 163 | 200 | 193 | | <130 |
| Triglycerides | mg/dL | 174 | 256 | 351 | | <200 |
| CMV-DNA | copies/mL | | 0 | 0 | | 0 |
| EBV-DNA | copies/mL | | 0 | | | 0 |
| Sirolimus | ng/mL | | 8.91 | 8.40 | 1.16 ** | |

After the mother noticed initial uterine contractions, sirolimus was stopped at 38 GW and labor was induced by dinoproston (Minprostin, Pfizer Pharma Gmbh) at 39 weeks and 1 day of gestation. The delivery was uncomplicated (Apgar 8/9/10 at 1, 5, and 10 min) and the newborn girl had a normal length (51 cm = 45th percentile), weight (3120 g = 28th percentile), and head circumference (33.5 cm = 19th percentile) for her gestational age. Blood values, including IL6, CRP, hemoglobin, leukocytes, leukocyte differentiation, thrombocytes, and CD4/CD8 ratio, were in the normal range. Only natural killer lymphocytes were initially below the normal range for her age, as in the mother before treatment, but rose to normal levels by the fifth day of life (Table 2). Sirolimus level obtained from cord blood was still 1.59 ng/mL. Cerebral MRI on day 6 of life confirmed subependymal nodules (Figure 2) and revealed a left frontal tuber and a small subdural hematoma probably caused by delivery.

**Table 2.** Selected laboratory values of the newborn. * from cord blood, 8 days after terminating sirolimus treatment; ** at postnatal day 1; *** at postnatal day 5.

| Parameter | Unit | Postnatal Day 1 | Postnatal Day 3 | Postnatal Day 5 | Reference Values |
|---|---|---|---|---|---|
| CD4 cells | /nL | | 1.66 | 1.96 | 0.40–3.50 |
| CD8 cells | /nL | | 0.57 | 0.66 | 0.20–1.90 |
| CD4/CD8 | ratio | | 2.9 | 3.0 | 1.0–2.6 |
| NK cells | /nL | | 0.05 | 0.12 | 0.10–1.90 |
| CRP | mg/L | <0.6 | | | <5.0 |
| IL 6 | ng/L | 2.2 | | | <66.4 |
| Hemoglobin | g/dL | 18.2 | | 19.0 | 13.2–21.7 |
| Erythrocytes | /pL | 4.8 | | 5.3 | 3.9–6.3 |
| Leukocytes | /nL | 14.55 | | 6.95 | 9.00–28.20 ** / 7.20–21.60 *** |
| Thrombocytes | /nL | 238 | | 2.52 | 220–520 |
| Sirolimus | ng/mL | 1.59 * | | | |

After discovering epileptiform discharges in the EEG, antiseizure medication was started with vigabatrin (Kigabeq 100 mg tablets, Orphelia Pharma SAS), according to the recommendations at present [19]. Echocardiography performed postnatally (Figure 3) showed only the partial covering of the tricuspid valve without inflow obstruction, so that cardiac surgery could be avoided, and a decision was made in favor of further treatment with a rapamycin analog. Here, everolimus (Certican® 0.1 mg suspendable tablets, Novartis

Pharma GmbH) was chosen because of the experience with this drug in newborns [20] and its easier administration compared to sirolimus.

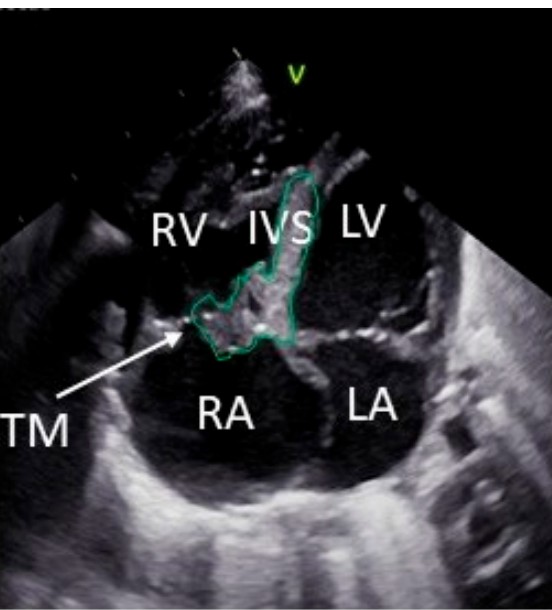

**Figure 3.** Echocardiografic four-chamber view at date of birth showing reduction in the right ventricular tumor size (tumor circumference in green). IVS: interventricular septum; LA: left atrium; LV: left ventricle; RA: right atrium; RV: right ventricle; TM: tumor.

Because of asymptomatic ventricular preexcitation, betablocker therapy (metoprolol tartrate 0.1 mg/mL solution; in-house production) was also started. After a further eight weeks, the right atrial tumor disappeared, as well as the left ventricular posterior wall rhabdomyoma. In the intraventricular septum, the size of the tumor decreased from 82% to 35%, measured as the ratio of the largest tumor size (LTS)/right ventricular diameter + LTS in the four-chamber view (Figure 4).

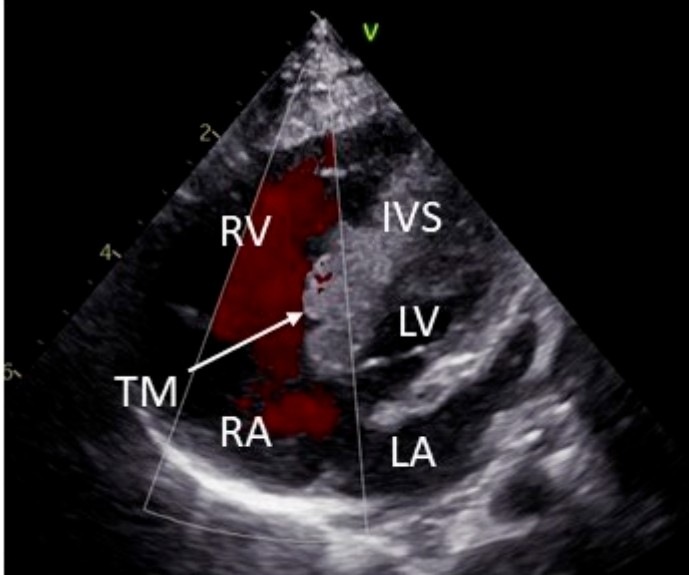

**Figure 4.** Echocardiografic four-chamber view at age of 2 months showing further reduction in the right ventricular tumor size during specific therapy. The tricuspid valve is tumor-free. IVS: interventricular septum; LA: left atrium; LV: left ventricle; RA: right atrium; RV: right ventricle; TM: tumor.

During her first two years of life, the child had a normal growth (height and weight at 17th percentile for age). A further echocardiography at the age of 13 months revealed a 100% reduction in tumor mass in the interventricular septum and only a small residual rhabdomyoma (3 mm) at the lateral base of the tricuspid valve with a °II insufficiency.

Her motor and mental development was retarded: at 13 months of age, she rated 55 points on the motor scale of the Bayley Scales of Infant Development (3. Ed.); she achieved unaided walking only at the age of 19 months. With respect to the more severe TSC phenotype of patients with *TSC2* mutations, compared to *TSC1* [21,22], this could be expected and speaks against a beneficial effect of the treatment with rapalogs for neurodevelopment in this case. However, to date, there have been no clinical signs of an autistic spectrum disorder. She only had few complex partial seizures and is, at present, seizure-free with the ongoing anti-seizure medication of vigabatrin.

Since the age of 12 months, she had several bouts of bronchitis and pneumonia, possibly caused by gastroesohageal reflux. The immunoglobulins G, M, and A in her blood were in the normal range, and primary ciliary dyskinesia as well as cystic fibrosis were ruled out. After pneumonia caused by a parainfluenza virus infection with bacterial superinfection at the age of 20 months, everolimus was stopped.

### 3. Discussion

Large rhabdomyomas hampering the heart function may pose a rare yet severe problem in fetuses and children with TSC. In their population-based study, Webb et al. reported on 15 children with symptomatic cardiac rhabdomyomas (12 with clinically assured TSC) [23]. One of these died prenatally of hydrops fetalis, while four died of heart failure in the neonatal period, with left-ventricle outflow tract obstruction (2), or "huge intramural tumors" (3), and/or arrhythmia (3). This finding of a high mortality rate was possibly due to the study design. However, Jozwiak et al. [8] found four cases of heart failure in their group of 42 children < 2 years with rhabdomyomas. Shi et al. reported that 14 out of 100 children with rhabdomyomas had to undergo surgery at an age < 3 months, three of whom died perioperatively; another child died of cardiac failure on the fourth day of life [3].

Treating the mother with a rapamycin analog, such as sirolimus or everolimus, offers an opportunity to prevent heart failure in the fetus or newborn. There have been six reports, including eight cases, to date (see supplement, Table S1). The reason to decide on treatment was progressive rhabdomyoma growth and impending heart failure in all reported cases [10,11,13–15] except one [12], in which the main cause was lymphangioleiomyomatosis in the mother that had already been treated with sirolimus before the pregnancy. In another case [15], the mother received sirolimus for renal angiomyolipomas before becoming pregnant. The decision to treat was usually made in the third trimester of pregnancy, when spontaneous tumor regression with the amelioration of cardiac function could not be expected any more. In all cases, the diagnosis of TSC was confirmed clinically [19] or by genetic analysis. Despite diverse treatment regimens with sirolimus doses varying between 3 and 12 mg daily, and quite varying maternal serum trough levels (Table S1), the treatment outcome was satisfactory in all reported cases. It should be noted that treatment had to be resumed in one case in the neonatal period because of recurrent tumor growth [10]. The levels of sirolimus measured in maternal serum and cord blood after delivery, as reported in the literature [10,12] and in this study, seem to indicate that the sirolimus clearance was slower in the fetus compared to the mother. As treatment benefit must be weighed against possible risks, the lowest effective dose should be chosen. In early pregnancy, rapalogs are considered to be relatively contra-indicated and should only be used where urgently needed, e.g., in transplant patients [24]. In the late-gestation period, the treatment of mice (however, not *TSC1/2* knockout mice) with rapamycin led to pups with a lower birth weight and reduced left ventricular mass, which persisted into adulthood [25]. However, there has been only one report of growth restriction in a fetus treated with sirolimus during the late-gestation period, associated with hypertension and proteinuria in the mother [11]. Because of the immunosuppressive effect of rapalogs [22], there may be a risk for a pre- or postnatal

infection in the mother or child. In the cases of prenatal sirolimus treatment [10–15], no severe infections were reported. In our case, we observed low natural killer cells in the mother before treatment as well as in the newborn postnatally. Apart from the association with combined variable immunodeficiency, and as an isolated finding, as measured in this newborn, low natural killer cells do not seem to bear a significant risk for severe infections [26]. Moreover, lower-airway infections in the child started only after the first year of life and were therefore probably not related to the prenatal rapalog treatment.

Other possible adverse effects of the rapalogs include thrombocytopenia and the perturbation of wound healing, which may cause birth complications. However, none of these possible adverse effects of a rapalog treatment in late gestation have been reported to date. The small subdural hematoma observed in this case was very likely not related to the sirolimus medication, as there was no thrombocytopenia in the newborn, and birth-related subdural hematomas are a common finding [27].

Here, we report another case of a successful treatment of a large rhabdomyoma in a fetus with TSC with sirolimus administered to the mother. The treatment effect measured by the tumor's size reduction was comparable to that reported in the literature [11,13,15]. With interdisciplinary management, heart failure in the fetus as well as possibly risky postnatal cardiac surgery could be prevented, although the chosen dose and effectuated serum levels of sirolimus were a slightly lower than that reported by others. We noticed only minor, reversible side effects of the treatment in the mother and none in the fetus and newborn, respectively. The somatic development of the child under ongoing treatment with everolimus was satisfactory.

Finally, this case report sheds light on the topic of an optional effect of early mTOR inhibitor treatment on epilepsy and neurodevelopment in TSC. Consistent with the constellation of an early clinical manifestation against the background of a *TSC2* mutation [22], neurodevelopment seems to be impaired in this case, in spite of prenatal treatment with a rapamycin analog. In addition, epilepsy developed in spite of this treatment, which was sufficiently controlled with vigabatrin as a specific anticonvulsant [18]. Controlled studies on this issue of early or preventive treatment with mTOR inhibitors in TSC, including the long-term outcomes for children, are desirable, but probably require a multicentric approach to attain sufficient numbers necessary for a reliable result.

## 4. Conclusions

We presented another example of the successful treatment of a life-threatening cardiac rhabdomyoma in a fetus with TSC by administering a rapalog to the pregnant mother. With sirolimus blood concentrations not exceeding 10 ng/mL (trough) and 15 ng/mL (peak), no major adverse events occurred, either in the child or in the mother. However, more data on the efficacy and safety of the treatment of fetuses with rapalogs, especially on the possible long-term effects, are needed.

**Supplementary Materials:** The following supporting information can be downloaded at: https://www.mdpi.com/article/10.3390/pediatric15010020/s1, Table S1: Cases reported on prenatal sirolimus treatment for fetal rhabdomyoma.

**Author Contributions:** Conceptualization, J.C.W., N.S., C.H. and B.W.; acquisition of data, J.C.W., N.S., T.M.G., O.S., W.H. and B.W.; analysis and interpretation of data, J.C.W., W.H., C.H. and B.W.; drafting, J.C.W. and B.W.; revisions, all authors; final approval, all authors. All authors have read and agreed to the published version of the manuscript.

**Funding:** This research received no external funding.

**Institutional Review Board Statement:** The study was conducted in accordance with the Declaration of Helsinki. According to the local legal regulations, an EC or IRB statement was not necessary in this case of an individual treatment decision.

**Informed Consent Statement:** Informed consent was obtained from all subjects involved in the study, i.e., the pregnant mother as well as the father of the unborn child. Treatment was subsequently begun off-label.

**Data Availability Statement:** Not applicable.

**Acknowledgments:** The authors thank D. Ebrahimi-Fakhari/Münster and H. von Bernuth/Berlin for their comments and recommendations. The authors thank the parents for providing their consent to the publication of personal information contained in this article.

**Conflicts of Interest:** The authors declare no conflict of interest.

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
