# Peer review of "Successful Prenatal Treatment of Cardiac Rhabdomyoma in a Fetus with Tuberous Sclerosis"

_pediatrrep, doi:10.3390/pediatric15010020_

Round 1

Reviewer 1 Report (Previous Reviewer 1)

Dear authors
Thank you for considering my previous comments. I believe it is valuable manuscript . It brings new knowledge and experience to the treatment of patients with rare diseases, such as heart tumors.

I recommend the manuscript for publication without further explanation and
revision.
Sincerely, Reviewer

Author Response

Dear Reviewer,

thank you for your kind comment.

Sincerely,

Bernhard Weschke on behalf of the authors

Reviewer 2 Report (Previous Reviewer 2)

I read with interest the case report entitled "Successful prenatal treatment of cardiac rhabdomyoma in a fetus with tuberous sclerosis"

The summary is well-conceived, but don't use the term "baby". Use more professional expressions.

Do not mix present and past tense. Use the past tense in all sentences.

The introduction is generally nicely written, but it might be convenient to add a few more pieces of information in which indications sirolimus and everolimus are used today, except for the ones mentioned.

I ask that the figures in the manuscript appear in the order they are listed in the text. Figure 1 must be followed by figure 2, not 4. Like figure 1, and with figures 2 and 3, add the meaning of all the abbreviations from the figure.

Please write in percentages how much further reduction of the rhabdomyoma was for the child's age of 13 months.

Table 3 is nicely designed, but I suggest that, for the sake of the size and clarity of the manuscript, you add it to the supplementary material.

Could you be more careful when you express yourself with the sentence "This high mortality is probably due to the study design."

Be careful with the phrase "Ethical review and approval were waived due to the urgent need for treatment in the situation of impending heart failure in the fetus"

Write references in accordance with the guidelines of the journal. I don't see the reference under number 21 in the text.

The manuscript is well written, but please take into account the previous comments, according to which you should once again read your own manuscript in detail and make the necessary changes.

Author Response

Dear Reviewer,

thank you for your comments and suggestions. Please find our responses in the attachment.

Sincerely,

Bernhard Weschke on behalf of the authors

Round 2

Reviewer 2 Report (Previous Reviewer 2)

Thank you for the answers. I am of the opinion that the manuscript in this form is acceptable for publication.

This manuscript is a resubmission of an earlier submission. The following is a list of the peer review reports and author responses from that submission.

Round 1

Reviewer 1 Report

This is an interesting case report on the treatment of a fetal heart tumor and deserves publication. However, there are some issues that require comment or clarification.

1. Heart tumors are most often the first manifestation of tuberous sclerosis and are present in fetuses. • Moreover, there are reports of spontaneous reduction or complete regression of the huge or multiple tumors - without the Sirolimus treatment. This requires a comment. Indicate please the frequency of rhabdomyoma among other cardiac tumors in children. (reference. e.g.: Echocardiography and Newer Imaging Techniques in Diagnosis and Long-Term Follow-Up of Primary Heart Tumors in Children. Int J Environ Res Public Health. 2020 Aug; 17(15): 5471. Published online 2020 Jul 29. doi: 10.3390/ijerph17155471)

2. • The authors describe that the indication for the use of SIROLIMUS was impending heart failure. The symptoms in fetus were like in advanced heart failure - fluid in the pericardium, decreased contractility, cardiomegaly, tricuspid regurgitation (Fig 1). Were there any other symptoms of heart failure on fetal ultrasound or  arrhythmia  ? This should be corrected.

3. • The first ultrasound was at 21 weeks, while the attached Fig.1  at 25 weeks. What was the progression of tumor growth during this time? When did the symptoms of heart failure appear? 

4. • The therapy of fetal cardiac rhabdomyomas  by sirolimus was started at 27 weeks. Did the authors consider the treatment earlier?

5.  Most tumors regress spontaneously, therefore the fact should be clearly emphasized indication to treatment of cardiac rhabdomyoma are meaningful cardiac failure and a life-threatening condition.

Reviewer 2 Report

Please correct the title. I suggest the following "Successful prenatal treatment of cardiac rhabdomyoma in a fetus with tuberous sclerosis"

An abstract is too short and written too generally. Please focus more specifically on your case.

For keywords, please try to use "MeSH" terms.

At the end of the introduction, write briefly what the aim of the case presentation, and how many cases have been described so far in the world.

It is not necessary to write "who had given birth two years ago" in the case description itself.

More information is needed about the mother's previous illnesses if she had them, possible medications, performed other prenatal diagnostics (e.g. NIPT), etc...

Please provide the figures in a better resolution and mark the left atrium on them.

Please indicate the size of the tumor (Figure 1).

Please submit the MR image and the 3D reconstruction.

In the case description itself, it is unnecessary to use "...and reports of successful intrauterine treatment of fetal cardiac rhabdomyomas..."

Please provide a reference with the recommendation (recommendation: D.Ebrahimi Fakhari/Münster).

With the drug sirolimus, you must specify the manufacturer. The same applies to all the drugs used in the case description.

When describing the values you monitored in the mother, it would be useful to form a table with which it would be easier for readers to follow the values throughout the weeks of treatment. Also, list the referral values you have followed.

You must also specify the manufacturer of the ultrasound with which the images were taken.

Please avoid the terms "normal length and weight for gestational age". Everything you state, specify exactly.

As with the mother, list all the values of the monitored parameters in the newborn in the table along with the reference ranges.

Also, attach a figure of the cerebral MR.

With figure 2, also mark and state the size.

At the first mention of a beta blocker, immediately state which beta blocker was used.

In the discussion, you must definitely touch on the literature review in more detail and show in one table all the patients treated so far (either prenatally or postnatally) with the mentioned drugs.

It is not advisable to use interrogative sentences in the discussion. Please revise them.

For the stated reason, it is necessary to revise the discussion in detail, which will clearly compare its own results with previous studies.

In the discussion and in the introduction, you did not touch on a number of works that are of interest;

- Webb DW, Thomas RD, Osborne JP. Cardiac rhabdomyomas and their association with tuberous sclerosis. Arch Dis Child. 1993 Mar;68(3):367-70.

- Gamzu R, Achiron R, Hegesh J, Weiner E, Tepper R, Nir A, Rabinowitz R, Auslander R, Yagel S, Zalel Y, Zimmer E. Evaluating the risk of tuberous sclerosis in cases with prenatal diagnosis of cardiac rhabdomyoma. Prenat Diagn. 2002 Nov;22(11):1044-7.

- Armada RC, Longchong Ramos M, Marrero P, Pascual J. Embryonal rhabdomyosarcoma associated with tuberous sclerosis. Med Pediatr Oncol. 2002 Apr;38(4):302.

- Wu SS, Collins MH, de Chadarévian JP. Study of the regression process in cardiac rhabdomyomas. Pediatrician Dev Pathol. 2002 Jan-Feb;5(1):29-36.

- Wan X, Harkavy B, Shen N, Grohar P, Helman LJ. Rapamycin induces feedback activation of Akt signaling through an IGF-1R-dependent mechanism. Oncogenes. 2007 Mar 22;26(13):1932-40.

etc...

Please cite all references in accordance with the instructions for authors.